# Light-Weight Wearable Gyroscopic Actuators Can Modulate Balance Performance and Gait Characteristics: A Proof-of-Concept Study

**DOI:** 10.3390/healthcare11212841

**Published:** 2023-10-27

**Authors:** Bram T. Sterke, Katherine L. Poggensee, Gerard M. Ribbers, Daniel Lemus, Heike Vallery

**Affiliations:** 1Department of Rehabilitation Medicine, Erasmus Medical Center, 3015 GD Rotterdam, The Netherlands; k.poggensee@tudelft.nl (K.L.P.); g.ribbers@erasmusmc.nl (G.M.R.); h.vallery@irt.rwth-aachen.de (H.V.); 2Faculty of Mechanical, Maritime and Materials Engineering, Delft University of Technology, Mekelweg 2, 2628 CD Delft, The Netherlands; dslemusp@gmail.com; 3Rijndam Revalidatie, Westersingel 300, 3015 LJ Rotterdam, The Netherlands; 4Faculty of Mechanical Engineering, Rhine-Westphalia Technical University of Aachen, 52062 Aachen, Germany

**Keywords:** wearable robotics, balance, walking, control moment gyroscope, postural control, falling

## Abstract

Falling is a major cause of morbidity, and is often caused by a decrease in postural stability. A key component of postural stability is whole-body centroidal angular momentum, which can be influenced by control moment gyroscopes. In this proof-of-concept study, we explore the influence of our wearable robotic gyroscopic actuator “GyroPack” on the balance performance and gait characteristics of non-impaired individuals (seven female/eight male, 30 ± 7 years, 68.8 ± 8.4 kg). Participants performed a series of balance and walking tasks with and without wearing the GyroPack. The device displayed various control modes, which were hypothesised to positively, negatively, or neutrally impact postural control. When configured as a damper, the GyroPack increased mediolateral standing time and walking distance, on a balance beam, and decreased trunk angular velocity variability, while walking on a treadmill. When configured as a negative damper, both peak trunk angular rate and trunk angular velocity variability increased during treadmill walking. This exploratory study shows that gyroscopic actuators can influence balance and gait kinematics. Our results mirror the findings of our earlier studies; though, with more than 50% mass reduction of the device, practical and clinical applicability now appears within reach.

## 1. Introduction

Falling is a major cause of morbidity, with an increasing prevalence in society due to its correlation with age [1,2]. Falls can occur during dynamic tasks, such as walking, and are often related to a decrease in balance and gait stability [3]. Impaired balance and mobility can be caused by sensory deficits (e.g., reduced vision, peripheral neuropathy, vestibulopathy), neurodegenerative disorders impacting movement control (e.g., Parkinsonian syndrome, cerebellar ataxia, vascular encephalopathy), and functional disorders like a fear of falling [4].

Many definitions exist for gait stability and balance [5]. Here, we refer to balance as the person’s ability to maintain the body’s centre of mass (COM), on average, within the base of support [6,7]. We refer to gait stability as the ability to keep walking despite the presence of control errors and disturbances [8]. Both describe the person’s ability to prevent falling. A differentiation can be made between global stability, i.e., a person’s susceptibility to falling, and local stability, i.e., step-to-step stability related to deviations from nominal gait [9]. As it is hard to directly affect global stability with robotic devices, one needs to resort to the more directly measurable local stability measures. For local stability, a plethora of outcome measures exist [3,10,11], such as kinematic variability [3], Lyapunov exponents [12], the extrapolated centre of mass [13], and whole-body centroidal angular momentum (H) [14,15].

Robotic devices have the potential to improve balance and gait stability by exerting forces and moments onto the body [16,17]. This helps improve balance, either through training or by direct assistance. A wide array of (robotic) assistive devices exists, ranging from a simple cane to full-body exoskeletons, each with their own influence on posture, e.g., by providing vertical unloading forces and body weight support [18], mediolateral pelvis stabilisation [19], or support via the hand [20].

Another method of influencing postural control is by manipulating H directly [14,15]. Angular momentum H has been used in robotics applications, for example, as a constraint for the control of bipedal robots like Boston Dynamics’s Atlas [21]. Several wearable methods for influencing H are currently being explored. One method relies on cold-gas thrusters, which employ compressed gas flowing through a nozzle. A reaction force is produced through the rate of change in momentum of the gas [22]. Finn-Henry et al. (2023) showed in a preliminary study that the likelihood of a unidirectional fall could be reduced with the use of such thrusters [23]. Another possible method to adjust H is the use of angular momentum exchange devices containing one or more flywheels, such as a control moment gyroscope (CMG). CMGs can generate moments without anchoring to an inertial frame making them excellent candidates for wearable devices to influence H [24,25,26,27,28].

Previous studies that focused on wearable devices containing CMGs have generated promising results [24,28,29]. Lemus et al. (2020) found that by exerting moments onto the upper body, non-impaired and post-stroke individuals performed better on various balance tasks. However, existing devices are too heavy (15 kg, Romtrairat et al., 2020; 16 kg, Lemus et al., 2020) to be viable for wearable applications [25,28]. In order to investigate whether more lightweight and energy-efficient CMGs can still influence balance and gait, we have developed a 4.9 kg backpack (GyroPack) containing two small CMGs [30].

In previous studies, we showed that the type of controller, and the resulting torque profiles, influence balance performance [28]. To focus our present study, we selected the best-performing controller type from previous studies: a rotational damper [28]. From other human–robot interaction studies, it is known that scaling of stiffness and damping forces can change a device’s impact on gait kinematics [31,32] and that optimal settings can vary across individuals [33,34]. Therefore, we explore two assistive dampers, with high and low damping coefficients. Additionally, we investigate one disruptive damper, with a negative damping coefficient, which could allow applications as a training device, following concepts of error-augmentation [35,36].

In this proof-of-concept study, we focus on non-impaired participants. We opted for a mixture of challenging static and dynamic balance tasks because both are regularly used to investigate the effect of robotics on balance and gait stability [28,37,38,39]. For balance tasks, we focus on temporal performance outcome measures, as they resemble those used in clinical settings [40]. For the walking task, we focus on the kinematic variability of step width and trunk angular velocity, as they are indicative of gait stability [3,19,41]. Additionally, trunk angular velocity is a major contributor to H [42] and will, therefore, allow us to infer our device’s impact on the body’s overall angular momentum.

Some studies suggest that positive and negative performance expectations might influence objective and subjective postural stability [43], though opposing results are also found [44]. To correct for a possible placebo effect, we compared all device conditions to a placebo condition.

The primary objective of this proof-of-concept study is to explore how well balance and gait can be influenced by more lightweight, less powerful gyroscopic actuators. Two research hypotheses have been formulated accordingly: First, we hypothesise that our device, with small gyroscopic actuators, is able to enhance performance on beam balance, both for the low- and high-intensity assistive controllers. Second, we hypothesise that gait stability, measured by a reduction in step width variability and trunk angular velocity variability, can be improved by using the damping assistive controllers.

Furthermore, we expect that there will be no significant differences in balance performance and walking characteristics between a “placebo” mode compared to baseline conditions. The secondary objectives of this research include investigating the effects of a negative damper, the presence of a learning effect in task performance, exploring personal preferences among participants, and determining whether demographic factors influence the response to the GyroPack.

## 2. Materials and Methods

After signing informed consent forms, a convenience sample of 15 participants (sex: 7 female, 8 male, age: 30 ± 7 years, weight: 68.8 ± 8.4 kg, and height: 174 ± 7 cm) were included in this exploratory study. The study was preregistered online (31 May 2022) (https://osf.io/yh8wm/) after approval by the Human Research Ethics Committee (HREC) of the Technical University of Delft (ID: 2136, 13 April 2022). The following inclusion criteria were used: (1) age between 18 and 55, (2) weight up to 150 kg, (3) must be able to fit the device, (4) must be able to tolerate upright standing and walking with device of 4.9 kg for 2 h. Participants were excluded in case of: (1) a self-reported history of balance impairments, (2) current medical issues that impede full weight bearing or ambulation, (3) self-reported pregnancy.

The GyroPack is a 4.9 kg backpack which contains two CMGs, each able to impart 15 Nm for 0.1 s in a fixed direction before reaching singularity. A change in desired torque direction re-enables the CMGs to impart a moment. The individual CMGs are described by Meijneke et al. (2021) [30]. Both CMGs are mounted with their gimbal axis aligned with the longitudinal axis of the trunk, see Figure 1. The device is able to impart a moment in the frontal and sagittal planes. All controllers used an inertial measurement unit (IMU) located within the backpack to obtain the trunk angular velocity vector (ω), with components in direction of ${\hat{u}}_{u}$ (frontal, roll), ${\hat{u}}_{v}$ (sagittal, pitch), and ${\hat{u}}_{w}$ (transversal, yaw). The third component of angular velocity, yaw rotation about the upright axis, was ignored in all controllers.

Four controller settings were developed: First, in the *placebo* mode, the angular momentum vectors of both CMGs were maintained at relative 180°, such that their effects cancelled each other. To mimic the motor noises produced by the other controllers, the gimbal velocity was coupled to the magnitude of ω around the frontal and sagittal axes, with a small gain. As both gimbals opposed each other, the CMGs combined produced a negligible moment. Second, in the *low* mode, a positive rotational damper was implemented. In this mode, the CMGs impart a moment proportional and opposite to the transverse-plane angular velocity vector ω. By opposing the trunk’s velocity, the CMG dampens its motion. Third, in the *high* mode, an identical rotational damper was implemented as in the *low* mode, and the gain relating ω and the gimbal velocity was twice as high compared to the *low* mode. This setting made the CMGs exchange their available angular momentum about twice as fast, intensifying the damping behaviour. The same amount of angular momentum can be exchanged, but the magnitude of the moment peak differs. Fourth, in the *negative* mode, a negative rotational damper was implemented. In this mode, the CMGs also imparted a moment proportional to ω, but acting in the same direction. In other words, it tends to amplify angular velocity, which can also be seen as error augmentation [45]. The backpack was tethered to a base station for computing and power supply. A full description of the device and controllers can be found in the Appendix A.

During all experiments, participants were attached to the ceiling via a safety harness to prevent falling. Motion capture cameras (Oqus/Miqus, Qualisys AB, Gothenburg, Sweden) captured kinematics. Fourteen markers were placed on each participant before the trials: C7—7th cervical vertebra, STRN—sternum, LSHO—left shoulder, RSHO—right shoulder, LPSI and RPSI—left and right posterior superior iliac spine, LASI and RASI—left and right anterior superior iliac spine, LANK and RANK—left and right lateral malleolus, LHEE and RHEE—left and right heel, LTOE and RTOE—left and right 1st metatarsal. Five markers were permanently attached to the GyroPack. Pre-processing of the data, i.e., marker labelling, and event detection was done in QTM (Qualisys AB, Sweden).

The protocol consisted of three parts: starting with static and dynamic balance tasks on an overground *balance beam*; followed by normal walking on a *treadmill*; and finally a qualitative questionnaire. Figure 1 shows the two experimental setups. All tasks were executed three times, under the following conditions (also see Figure 2):**Pre-baseline:** baseline, no device donned;**Unpowered:** device donned, but turned OFF, i.e., not powered;device ON conditions (fully randomised):
(a)**Placebo:** placebo/sham;(b)**Negative:** negative damper;(c)**Low:** low-intensity damper;(d)**High:** high-intensity damper;**Unpowered:** device donned, but turned OFF, i.e., not powered;**Post-baseline:** baseline, no device donned.

**Figure 2 healthcare-11-02841-f002:**
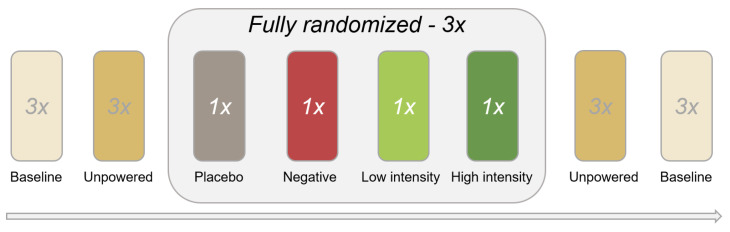
Visualisation of order of conditions applied during all balance beam tasks and treadmill walking.

The *balance beam* tasks were all performed on a 3 cm wide, 4 cm high, 500 cm long beam. Starting with static mediolateral (ML) standing balance, participants were asked to stand on the beam, for as long as possible, in a tandem stance (heel-to-toe). Subsequently, as a dynamic task, a tandem gait was performed across the beam. The distance was measured as the distance between the first and last correctly placed foot on the beam (heel-to-heel). During the last static balance task, subjects had to stand on the beam in the anteroposterior (AP) direction, i.e., feet shoulder-width apart and oriented perpendicular to the beam. All three tasks constituted one trial and were completed with the arms crossed across the chest, to minimise compensatory movements. The ML and AP standing time was measured as the time that both feet were out of contact with the floor, and were capped at 30 s, as this is regular practice in clinical tests, such as the mini-BESTest [46].

The *treadmill* task consisted of two minutes of walking at a fixed speed of 1.2 m/s on a wide treadmill without handrails (C-Mill, Forcelink B.V., Culemborg, The Netherlands). During walking, the arms were crossed across the chest to minimise the influence of arm swing on trunk kinematics [41]. The data of the last minute was used for analysis, allowing one minute adaptation time after device settings changed. Step width was calculated as the distance, perpendicular to the walking direction, between two subsequent heel strikes. Step width variability was calculated as the coefficient of variance of all step widths in the final minute. Trunk angular velocity (ω) was defined as the sum of the two angular velocity vectors *roll* (frontal plane rotation) and *pitch* (sagittal plane rotation) of the trunk segment with respect to the global inertial frame. The *yaw* (transversal plane rotation) was disregarded. To calculate the peak magnitude of ω for each stride, each stride cycle was divided into 100 frames. Subsequently, the average of each frame was calculated across all strides, which yielded the “average” stride trajectory for each trial. Of that average stride trajectory, the maximum of the magnitude of ω was taken as the peak trunk angular rate ($\omega_{\max}$). In order to calculate trunk angular velocity variability ($\omega_{\mathrm{SD}}$), at each frame of the average stride trajectory, the standard deviation between strides was calculated and subsequently averaged over the normalised stride cycle [47].

In addition to the biomechanical measures, after each trial, the participants were asked to provide a subjective *rating*, ranging from 0 to 10. The rating was based on the request: “Please rate the experience of the tasks, and setting of the backpack, between zero and ten, where normal walking on the street is a seven”. This semantic differential scale objectified the user’s experience of the usefulness of the device and helped assess the system’s usability [48].

In short, the primary outcome measures on the balance beam are: beam walking distance (in m), ML and AP standing time (in s), and subjective rating. On the treadmill, primary outcomes are step width variability, peak trunk angular rate, and trunk angular velocity variability.

To control for a possible placebo effect, we compared the paired condition means of the placebo condition versus the before and after baseline and unpowered conditions. To analyse the efficacy of the GyroPack, we then compared the differences in outcome measure means between the placebo condition and the device ON conditions (*negative*, *low*, *high*). Post hoc, we added a fourth condition, called *best*, containing the best-performing controller, for each participant separately, determined by the highest mean of either the *low* or *high* condition. To determine the presence of learning effects, we investigated the differences between the pre- and post-baseline measurements.

For all above-mentioned statistical analyses, a Wilcoxon signed rank test was used, with Bonferroni correction to account for multiple testing. We used an alpha level of 0.05 for all statistical tests. Additionally, the Wilcoxon effect size was calculated, which differentiates three levels to indicate small (<0.3), moderate (0.3–0.5), and large (>0.5) effect sizes. Lastly, the correlation between participant weight and difference in balance beam performance (placebo versus *low* or *high*) was investigated using a Spearman Rank test. Post-processing and statistical analysis were done using MATLAB™ (MathWorks, Natick, MA, USA) and R (R Foundation for Statistical Computing, Vienna, Austria).

## 3. Results

### 3.1. Placebo Effect

Figure 3 shows the task performances during the placebo vs. baseline conditions. The outcome means for the baseline is calculated by combining and averaging the pre- and post-baseline measurements for each participant. No significant differences were found when comparing the *placebo* versus the baseline or unpowered conditions. However, a significant increase in subjective rating was found, when comparing the *placebo* to the *unpowered* (*p* = 0.042, effect size = 0.650 (large)) and to the *baseline* (*p* = 0.026, effect size = 0.701 (large)) conditions.

### 3.2. Balance Beam

#### 3.2.1. Beam Walking

Figure 4 shows a box plot of the means per participant for the distances walked on the beam (in m) for the *placebo*, *negative*, *low*, *high*, and *best* conditions. Compared to the *placebo* condition, the mean distances were lower when using the *negative* damper (−23%), and higher with the *low* (+21%) and *high* damper (+36%), though not significantly. The selection of the *best* performances of *low* or *high* shows a significant increase of +44%, compared to *placebo* (*p* = 0.041, effect size = 0.645). The Spearman Rank test showed no significant correlation between the participant’s weight and the changes in balance beam performance. Post hoc inspection of weight versus best performing controller (*low* or *high*) provided no significant findings either.

#### 3.2.2. Mediolateral Standing

Figure 5 shows ML standing time across conditions. Performance on this task show similar non-significant differences as for beam walking, namely standing time was lower for *negative* (−17%) and higher for *low* (+32%), and *high* (+29%), compared to placebo. When considering the *best* performances, there is a significant increase in standing time of +50%, compared to placebo (*p* = 0.017, effect size = 0.704).

#### 3.2.3. Anteroposterior Standing

For the AP balancing trials, the same trends were visible, compared to *placebo*. Namely, the mean of *negative* was lower (−17%), while the means of the *low* (+18%), *high* (+6%), and *best* (+30%) were higher, compared to placebo. The results are plotted in Appendix B Figure A1. No significant changes were found.

#### 3.2.4. Subjective Rating

When looking at subjective rating of the condition, provided during the *balance* balance beam trials, the participants did significantly rate the *negative* condition lower, compared to the *placebo* condition (*p* = 0.029, effect size = 0.232). The other groups did not show a significantly different rating compared to *placebo* (see Figure 6). When comparing the rating of either the *low* or *high* controller to the distance walked on the beam, participants correctly identified their best performing controller in 77% of the cases. For two participants, the subjective rating was insufficiently recorded to include in the analysis and was considered missing data.

### 3.3. Treadmill Walking

The trunk angular velocity variability, $\omega_{\mathrm{SD}}$, changed significantly for all conditions. Specifically, variability decreased for the *low* (*p* = 0.021, effect size = 0.247 (small)) and *high* damper (*p* = 0.007, effect size = 0.212 (small)), while it increased for the negative damper (*p* = 0.004, effect size = 0.342 (moderate)). Figure 7 shows the variability of $\omega_{\mathrm{SD}}$ during aforementioned conditions.

During treadmill walking, a significant decrease in peak trunk angular rate, $\omega_{\max}$, was found for the *high* damper condition (*p* = 0.007, effect size = 0.177 (small)). Conversely, a significant increase was found for the *negative* condition (*p* = 0.037, effect size = 0.0707 (small)), when compared to placebo. For the *low* condition, no change was observed. The results are shown in Appendix B Figure A2.

No significant differences were found for step width variability across any of the trials.

For treadmill walking, the recordings of participants 1, 3, and 8 were disregarded due to poor marker visibility, resulting in loss of essential markers for calculation of the outcome measures.

### 3.4. Pre- and Post-Baseline

To investigate possible fatigue and learning effects, we compared pre- and post-baseline results on the balance beam tasks, shown in Figure 8. After correction for multiple measurements, a significant increase of 59% was found for the beam walking task (*p* = 0.018, effect size = 0.420 (moderate)). ML and AP standing time did not show significant differences.

## 4. Discussion

On the *balance beam*, neither the *low* nor the *high* condition individually achieved significance in increasing performance. However, when selecting each participant’s personal *best*-performing setting, the results on the beam walking and ML trials show that the GyroPack significantly improved static and dynamic balancing performance by +44% and 50% respectively, which relates to a large effect size (>0.6). The performance increase on the balance beam mirrors our earlier results with a much heavier device [28], while our current more lightweight wearable makes implementation in a clinical population more feasible.

On the *treadmill*, trunk angular velocity variability ($\omega_{\mathrm{SD}}$) significantly changed for all conditions. Specifically, variability increased as a result of the *negative* damper, and decreased as a result of the *low* and *high* intensity dampers. As $\omega_{\mathrm{SD}}$ is an indication of gait stability [47], these results show that the GyroPack can be used to modify the stability of steady-state walking.

Previous studies show that a rotational damper to the trunk can improve balance performance [25,28,37]. Trunk rotation and angular momentum damping is also used to control balance in bipedal robots [21,49]. Damping of other body parts, such as upper body [18] and pelvis [19], is similarly found to improve balance and gait stability. Our current findings further strengthen the hypothesis that damping of trunk rotation can improve balance and gait stability [10].

The GyroPack can also be used to challenge balance and stability. The negative damper, compared to placebo, shows a significant increase in peak trunk angular rate ($\omega_{\max}$) and trunk angular velocity variability ($\omega_{\mathrm{SD}}$), during walking on a treadmill. On the balance beam, non-significant trends are visible for the negative damper compared to the placebo condition (−25% distance on the beam and −17% ML and −18% AP standing time). These findings might prove relevant in balance training with “error augmentation,” a strategy that aims to enhance learning by increasing feedback error [36,50]. De Luca et al. (2020) showed that, following a robotic training protocol focused on balance and core stability, individuals post-stroke improved more on reactive balance and trunk control compared to controls [51]. Similar training protocols can be envisioned for the GyroPack.

In the field of wearable robotics, the user’s preference is emerging as a formally quantifiable outcome metric, capable of describing factors that are difficult to measure but are important to the user [52,53]. The differences in response to the *low*- and *high*-intensity dampers shows the importance of personalised robotic controllers. In 77% of the cases, participants were able to correctly identify their best performing controller. However, the true personal *best* controller for each participant might even have been somewhere in between, or outside, the settings of the preset *low* and *high* controllers.

The significant increase in subjective rating for the placebo conditions, when compared to baseline and unpowered, shows that the participants’ perception of the effect of robotic devices can change due to unrelated effects of a device that do not necessarily assist movement, such as sound. No objective increase in performance was found. These findings are similar to those of Horváth et al. (2023), who found an association between expectation and perceived performance, although their placebo intervention did *not* affect actual test performance [44]. However, other studies do report objective increases in postural stability as a result of a (medicinal) placebo effect [43]. In their review, De Bock et al. (2022) found that none of their included exoskeleton studies blinded participants or investigators, even though expectation can affect device assessment outcomes and increase variability [54,55]. This shows the importance of adding such considerations to human–robot interaction studies.

When looking at the before and after baseline conditions, an increase in performance was shown for the beam walking. This is most likely due to learning, as also shown by Domingo et al. (2009) [56]. This shows the importance of performing pre- and post-baseline measurements. Especially when, in order to test the efficacy of a robotic wearable, difficult tasks are chosen to challenge non-impaired individuals.

### 4.1. Limitations

After the first two participants, it became clear that the originally registered protocol took too long to execute. Therefore, we dropped the mini-BEStest to reduce strain on the participants. We also reduced the walking time from three minutes to two minutes. It has been suggested that for an accurate measure of step width variability, at least 400 steps are required [57], i.e., about 10 min of walking. Due to the time constraint, this suggestion could not be met, possibly influencing the accuracy of our step width variability measurement.

In the pre-registration, we stated that we would perform an ANOVA to analyse group-level interaction effects. This should have been its non-parametric alternative, the Friedman Test, as the normality assumption of the ANOVA could not properly be checked, due to the low number of participants. Nevertheless, we decided not to report group-level analyses, as these looked at all of the interactions, such as between the positive and negative dampers, which would have inflated our results, and shown significance for all tasks. As we are only interested in the interactions of the individual controllers versus the *placebo* condition, we directly performed the non-parametric Wilcoxon signed rank test to compare those specific differences, with corrections for multiple testing.

### 4.2. Future Work

The GyroPack is intended to become a fully untethered assistive device, which requires on-board control and power, i.e., an EtherCAT master and a battery. In the worst case, each CMG consumes about 43 W. Adding an EtherCAT master, such as the C6015-0020 (Beckhoff Automation GmbH, Verl, Germany), adds 400 g to the system, with 15 W additional power consumption. Using 5x3 3.6 V Lithium cells, with 2550 mAh capacity and a 20% rated discharge margin, would give an effective total discharge capacity of 110 Wh and a mass of 700 g. This could power the device for more than an hour. With a total weight of 6.0 kg, the GyroPack would be able to perform at maximum capability during a therapy session or a walk through the park. Future work will focus on further reductions in weight and power consumption.

All segments of the body contribute to balance, gait characteristics, and H [42]. The individual gyroscopic actuators (approx. 1.5 kg) can help study populations with altered lower and upper extremity function, for instance by: modifying cueing in people with Parkinson’s disease [58], modifying inertial properties of objects for grasping tasks [59], through telerehabilitation [60], or by assisting obstacle avoidance [61]. In the current study, arm swing was restricted to increase task difficulty and reduce its influence on H and trunk angular velocity variability [41]. However, to evaluate the effect of the GyroPack on natural walking, future research should allow arm swing.

We now use a cascaded control scheme for the CMGs, based on Valk et al. (2018) [62], that appears like a damper only within the narrow range before singularities are hit, at which point the moment drops suddenly. Other control schemes could be more effective. Within the current control architecture, customisation of the device could be tuned by hand, but for controllers with more parameters, algorithmic approaches will have to be used. Future work will focus on optimisation and personalisation of the GyroPack control schemes.

## Figures and Tables

**Figure 1 healthcare-11-02841-f001:**
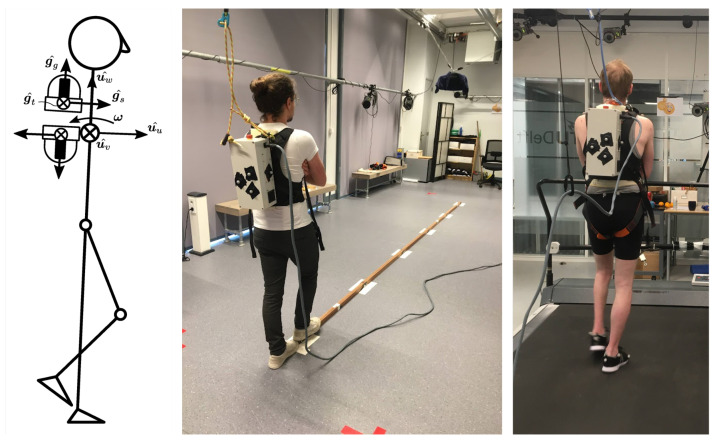
(**Left**): Stick-figure visualising the CMG on the back of the person, where ${\hat{g}}_{g}$, ${\hat{g}}_{s}$, and ${\hat{g}}_{t}$, respectively, denote the gimbal axis, flywheel axis, and the perpendicular output axis; ${\hat{u}}_{u}$, ${\hat{u}}_{v}$, and ${\hat{u}}_{w}$ denote the unit axes of the body coordinate frame; and ω is the trunk angular velocity vector about all three axes. (**Middle**): overground balance beam laboratory setup, participants walk and stand on the beam with their arms crossed. (**Right**): a wide treadmill, without handrails on the side, participants walk with arms crossed.

**Figure 3 healthcare-11-02841-f003:**
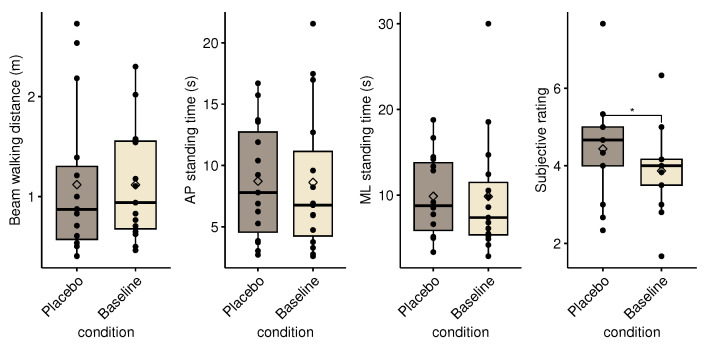
Box plot of performance outcomes and subjective rating during baseline and placebo conditions. The diamond marker shows the overall condition average. The black dots show the individual participant means. The asterisk denotes a significant difference between conditions (*p* < 0.05).

**Figure 4 healthcare-11-02841-f004:**
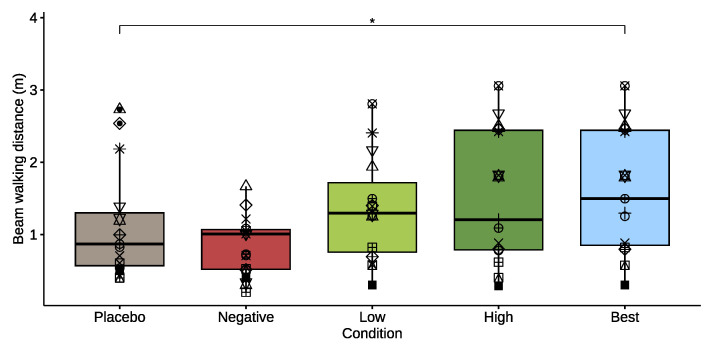
Box plot of distance walked on the beam for the *placebo* mode and for the *negative*, *low*, *high*, and *best* damper conditions. The asterisk denotes a significant difference between conditions (*p* < 0.05). Each participant is denoted by a different symbol to allow comparison across conditions.

**Figure 5 healthcare-11-02841-f005:**
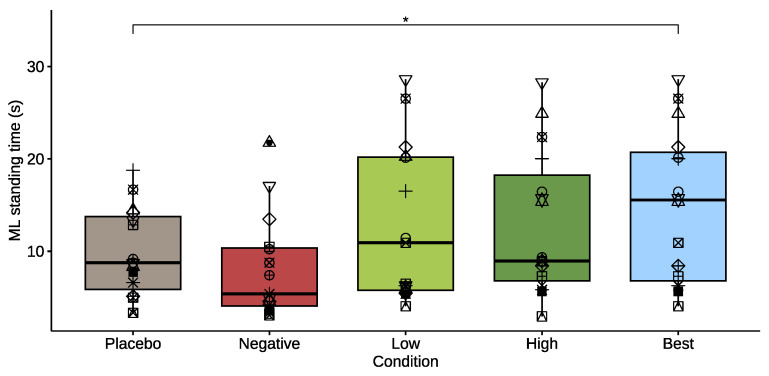
Box plot of mediolateral standing time for the *placebo* mode and for the *negative*, *low*, *high*, and *best* damper conditions. The asterisk denotes a significant difference between conditions (*p* < 0.05). Each participant is denoted by a different symbol to allow comparison across conditions.

**Figure 6 healthcare-11-02841-f006:**
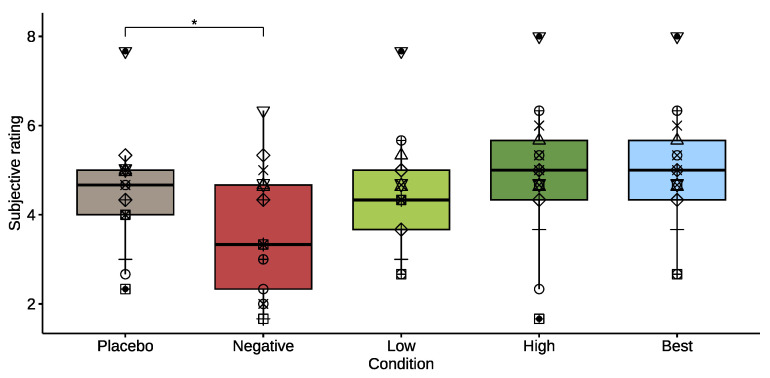
Box plot of the subjective rating of “how the condition feels compares to normal walking on the street”. Each participant is denoted by a different symbol to allow comparison across conditions. The asterisk denotes a significant difference between conditions (*p* < 0.05).

**Figure 7 healthcare-11-02841-f007:**
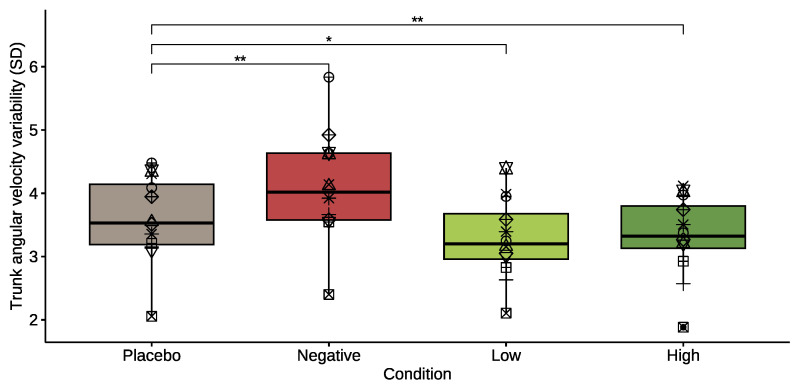
Box plot of the trunk angular velocity variability for the placebo mode and for the *negative*, *low*, and *high* damper conditions. Each participant is denoted by a different symbol to allow comparison across conditions. The asterisks denote significant differences between conditions (*—*p* < 0.05, **—*p* < 0.01).

**Figure 8 healthcare-11-02841-f008:**
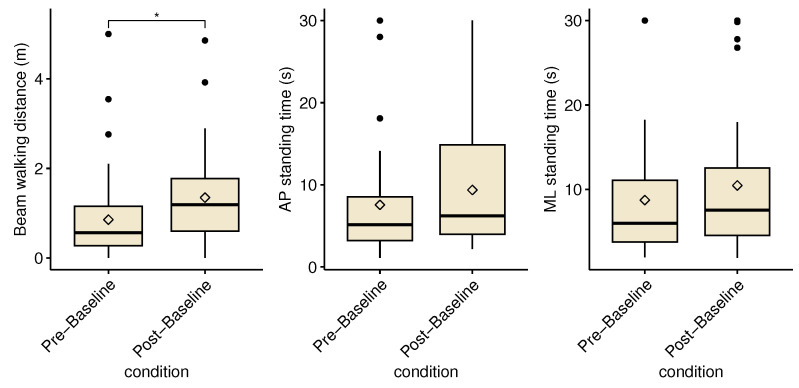
Box plot showing the pre- and post-baseline conditions for beam walking, and mediolateral and anteroposterior balancing. The diamond marker shows the overall condition average. The black dots show the individual participant means. The asterisk denotes a significant difference between conditions (*p* < 0.05).

## Data Availability

All generated data and associated scripts used for the study are published here: https://doi.org/10.4121/51cf7f2e-9cf6-4b4e-8944-7e7369514816.

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
