# Peer review of "Light-Weight Wearable Gyroscopic Actuators Can Modulate Balance Performance and Gait Characteristics: A Proof-of-Concept Study"

_healthcare, 2023, doi:10.3390/healthcare11212841_

Round 1

Reviewer 1 Report

Comments and Suggestions for Authors

The manuscript by Sterke and colleagues explore the utility of gyroscopic actuators in systematic perturbation of balance control and gait dynamics. The authors presented information on gyroscopic actuator in the supplementary document, and described its use on able-bodied participants. The experimental conditions are laid out well and I think the manuscript has potential to be published. However, there are several issues impending adequate clarification, statistical analyses, and potential applicability of the actuators to broader research and populations. I have highlighted my concerns below:  

1.       Lines 58-64: The structure of hypotheses needs to be modified so that the null hypothesis is not confused with research hypotheses (i.e., no effect between placebo and baseline conditions can be pitched as an expectation as opposed to a hypothesis).

2.       While the introduction leads the reader well to expect author’s proposed GyroPack to influence whole-body angular momentum (H), the description preceding hypothesis-2 does not dwell upon why GyroPack’s influence on H would enhance participants’ performance on beam balance – especially that of non-impaired participants for whom improvement in balance performance could be marginal. Adequate rationale as well as rewording of the hypothesis is essential here.

3.       Similarly, the information in the introduction is not adequate to support hypothesis-3. The effect of high and low-intensity damper and negative damper on balance control and gait needs to be clarified along with previous research in the field prior to leading the readers to the current hypothesis-3.

4.       The statistical testing of variables needs to incorporate group-level statistical analysis, such as ANOVA or its non-parametric equivalent Kruskal-Wallis test and assess pairwise comparisons via Wilcoxon tests provided the group-level statistics show significantly different p-statistic. The description as to why these tests were not conducted in the limitations section is inappropriate because non-parametric tests can be performed if normality assumption is an issue. The ANOVA or Kruskal-Wallis testing needs to be reported for variables including Beam walking distance, ML standing time, Subjective rating, and Trunk angular velocity variability.

5.       Potential applications of the gyroscopic actuators to upper extremity research and populations with altered lower and upper extremity function need to be contextualized for the readers. Especially in the discussion section, authors can highlight how the gyroscopic actuators could be incorporated to assess nuanced behavioral deficits in aging, patients with PD, and stroke to name a few, by referring to the following manuscripts:

a.       Ginis P, Nackaerts E, Nieuwboer A, Heremans E. Cueing for people with Parkinson's disease with freezing of gait: A narrative review of the state-of-the-art and novel perspectives. Ann Phys Rehabil Med. 2018 Nov;61(6):407-413. doi: 10.1016/j.rehab.2017.08.002.

b.       Mutsaarts M, Steenbergen B, Bekkering H. Anticipatory planning deficits and task context effects in hemiparetic cerebral palsy. Exp Brain Res. 2006 Jun;172(2):151-62. doi: 10.1007/s00221-005-0327-0.

c.       Rao N, Mehta N, Patel P, Parikh PJ. Effects of aging on conditional visuomotor learning for grasping and lifting eccentrically weighted objects. J Appl Physiol (1985). 2021 Sep 1;131(3):937-948. doi: 10.1152/japplphysiol.00932.2020.

d.       Chen SC, Lin CH, Su SW, Chang YT, Lai CH. Feasibility and effect of interactive telerehabilitation on balance in individuals with chronic stroke: a pilot study. J Neuroeng Rehabil. 2021 Apr 26;18(1):71. doi: 10.1186/s12984-021-00866-8.

Comments on the Quality of English Language

Proofreading the manuscript can help catching occasional typographical errors.

Reviewer 2 Report

Comments and Suggestions for Authors

For all authors

The manuscript is very interesting for researchers in special areas of robotics and biomechanics. The manuscript presented a very extensive study that could perhaps clarify some points for readers. line 5 people without disabilities - which participants were included; age, weight, women, men line 12 balance (parameters) and gait (parameters) line 157 further explain the question and the number of participants - method. The question had an influence on the results, and how Limitations: Why the study reduced walking time

Reviewer 3 Report

Comments and Suggestions for Authors

General comments:

The current study evaluates the effectiveness of a light weight control moment gyroscope in improving balance performance of healthy adults. Overall, the paper is well written, and the work put into the study is evident. The concept of a light weight device improving gait performance is interesting, but the paper currently lacks focus, and can be improved by including specific interpretations.

A major concern is that the discussions section lacks focus. There is a lack of specific interpretations related to the study.

a.      It would be best to focus on how the CMG effects/improves balance performance. Relate your results more to literature on balance performance. Currently the recommendations and interpretations are more generic, rather than directly related to the good work that you have done with the study.

b.     The importance of personalization pointed out in the first 2 paragraphs are generic recommendations.

c.      Lines 281 to 289 goes into the effects of placebo and psychological factors, which are generic, well-established and doesn’t add new information to the literature.

d.     The use of negative damper condition in balance training is interesting, and can be expanded on with more references to literature.

e.      Lines 261 to 262 if there are no significant results, it is better not to cherry pick a few comparisons and form an interpretation. Without significance, it could go either way. We just don’t know. Needs a citation for the clinical significance threshold stated.

Please define ML and AP standing time better in the methods. I am assuming it the time from beginning of balance test to when the subject loses balance.

Even though arm swings increase variability, I would argue that it is an important consideration in testing out any balance performance improving devices. Since all subject’s followed a consistent protocol, your study still helps us understand the effects of this light weight CMG on balance, but in the future, it would be better to allow for arm swings.

The results section has some interpretations (lines 187-188). It is best practice to present the results unbiased by opinions in the results section.

Round 2

Reviewer 3 Report

Comments and Suggestions for Authors

Thank you for taking the time to address all the comments. I believe the revisions has made the paper stronger. It is more clear and focused now.